# Identifying profiles of stressors and stress management strategies in Filipinos: A secondary analysis

Miguel Antonio Fudolig[1] 🔘, Pamela Paula Pioquinto[1], Marijo Villano[2] and Lorraine S. Evangelista[3]

[1]Department of Epidemiology and Biostatistics, School of Public Health, University of Nevada Las Vegas, Las Vegas, USA; [2]Counseling and Psychological Services, University of Nevada Las Vegas, Las Vegas, USA and [3]Sue & Bill Gross School of Nursing, University of California Irvine, Irvine, USA

## Research Article

**Keywords:**
stress; stress management; public health; latent class analysis; Filipino health

**Corresponding author:**
Miguel Antonio Fudolig;
Email: miguel.fudolig@unlv.edu

## Abstract

Chronic stress can lead to physical and mental health conditions. This study aimed to identify the different stress profiles and stress relief methods among Filipinos living in the Philippines using latent class analysis. A secondary analysis of a cross-sectional study was employed in this study. The stressors and stress-relief practices among Filipinos were investigated using the I-HEART-FILIPINOS data set. Latent class analysis was used to identify the different profiles of stress causes and management methods among 1,196 Filipinos residing in the Philippines, specifically the Northern Luzon area. Four stress-related profiles were identified: (1) low financial stress relieved by eating and exercise; (2) work-related stress relieved by self-care; (3) familial and economic turmoil relieved by eating, exercise and prayer; (4) high financial stress relieved by staying at home and remaining indoors. The four-class solution explained 58% of the variation in the data through classification. Disparities were observed between classes in terms of emotional distress and sociodemographic variables, implying how sociocultural factors could affect stress presentation and management in Filipinos. The findings of this study inform the development of stress management interventions specifically designed to address the needs of underserved populations in the Philippines and neighboring countries to improve overall health.

## Impact statement

This study employs a novel approach of latent class analysis to provide deep insight into the causes of stress among Filipinos and their coping mechanisms. Most studies on stress management and causes in Filipinos are specific subpopulations such as immigrants, university students and healthcare workers. The sample comprised rural and older adults, an understudied demographic in global mental health research. The findings revealed differences across various stressor and stress management profiles among Filipinos, highlighting the influence of sociocultural factors specific to the Philippines. For instance, prayer was found to be a major stress management practice across all the profiles. The reliance on prayer could be attributed to the country's colonial history and Catholic influence. Profiles that reported familial relations as a stressor were predominantly female, suggesting the influence of binary gender roles. The profile analysis employed in this study provides a reproducible and holistic framework for future research on stress presentation in rural and elderly populations in the Southeast Asian (SEA) region. These results provide a pre-pandemic baseline on stress management research of Filipinos and populations with similar demographics in other SEA nations. This study also offers evidence-based guidelines for tailoring stress management interventions to the Filipino population by characterizing local coping practices.





## Introduction

Stress, defined as a person's appraisal of their environment as having demands that exceed their ability to adapt or cope (Lazarus, 1984; Vancheri et al., 2022), is a growing concern worldwide (World Health Organization, 2011; Piao et al., 2024). Stressors are any stimuli that disrupt homeostasis and trigger a stress response, commonly referred to as "stress," which can range from personal factors to environmental challenges (McEwen and Stellar, 1993; Greenberg et al., 2002; Restrepo and Lemos, 2021; Guan et al., 2022). The transactional theory of stress and coping posits two phases of appraising stress. The first is the initial appraisal of the stressor to determine its degree of threat to the individual and the second is the appraisal of one's resources to cope effectively with the stressor (Lazarus, 1984; Biggs et al., 2017). The theory also states that both the appraisal and strategies employed may be dependent on situational and dispositional factors (Lazarus and Folkman, 1987). Although everyone experiences stress at some point in their

lifetime, excessive or chronic stress can lead to comorbid physical and mental health conditions, such as symptoms of anxiety, symptoms of depression, post-traumatic symptoms, substance use disorders and cardiovascular (CV) conditions, which includes coronary heart disease (CHD) or ischemic heart disease (IHD) (Kotlęga et al., 2016; Satyjeet et al., 2020; Vig et al., 2020; Balkan et al., 2022).

The epidemiology of stress is relatively widespread worldwide, but the burden is highest in low- and middle-income countries due to the unique challenges their populations face (Rathod et al., 2017). The association between perceived stress level and mental health outcomes was found to vary across different low- to middle-income countries. However, cultural context specific to the country may strengthen this association (Cristóbal-Narváez et al., 2020). While the prevalence of stress among Filipinos living in the Philippines is not well understood, socioeconomic challenges, such as poverty, contribute to the presentation of stress in this population (Mina, 2024). However, stress is not experienced uniformly, and certain groups within the population may experience different types of stress, such as workplace stress, financial stress and familial stress. Each group may also have distinct ways of coping with stress compared to others. In the Asian context, high pressure is prevalent in both work and academic settings, often stemming from poor working conditions, low wages and excessive workloads (Awang Idris et al., 2010; Dy et al., 2015a). Additionally, among Filipino students, academic pressure can also cause familial stress due to students' sense of indebtedness toward their families, indicating that family-related factors impact their well-being (Dizon et al., 2025).

Existing stress-related literature in Filipinos was focused only on specific subpopulations such as students, teachers, seafarers and urban individuals (Dy et al., 2015a; Tamanal et al., 2017; Bate et al., 2019; Yazon and Ang-Manaig, 2019; Serafica et al., 2023). In addition, there is a plethora of studies on the stressors of Filipino immigrants across the world (David and Nadal, 2013; Van der Ham et al., 2015; Yeung et al., 2021). However, these studies have left a significant gap in the literature regarding the presentation of stress in understudied Filipino subpopulations including rural, older adult Filipinos. In contrast, this study presents distinct phenotypes of stressors and stress-management typologies across different groups that includes representation from understudied populations such as rural and older adult Filipinos.

Pre-pandemic research on the emotional stress and mental health outcomes of South East Asian (SEA) populations focused on refugees (Beiser and Hou, 2001; Silove et al., 2017). During the COVID-19 pandemic, studies on mental health outcomes in South East Asia focused on the prevalence of stress and mental health disorders as the pandemic progressed (Balakrishnan et al., 2023). The data used in this study were collected in 2017, which was before the COVID-19 pandemic. The findings can serve as a baseline for post-pandemic studies related to the stress-related studies on mental health outcomes in the Filipino population and other similar populations in the SEA region. Building on global findings, a profile analysis of stress and stress relief methods is essential to better understand the diverse experiences of Filipinos living in different parts of the Philippines.

To address this gap, this study aims to identify and assess the sociodemographic correlates of different stress profiles and stress management strategies among Filipinos residing in the Northern Luzon region of the Philippines using latent class analysis (LCA). LCA is ideal for examining stress-related behavior patterns because it uses probabilistic methods to create a model-based grouping of heterogeneous populations based on outcome variables (Aflaki et al., 2022). Unlike distance-based clustering methods, LCA identifies patterns of observed categories across cases (Weller et al.,

2020). This study is the first of its kind to identify stressors and stress management behaviors within the Filipino community, which prior literature often analyzes separately. Additionally, this is the first study to use LCA in stress research in Filipinos. Recognizing different stress patterns stress management strategies before the pandemic, this research will contribute a valuable understanding of the differences within Philippine culture. These patterns can be extended to similar populations in SEA and low- to middle-income countries that share cultural similarities with the Philippines, which can enhance public health interventions at the regional level, specifically targeting stress as a risk factor for more severe health conditions.

## Methods

### Study design and setting

This study conducts a secondary analysis of the I-HELP-FILIPINOS data set, a community-based survey conducted in 2017 in Northern Luzon, Philippines, that examines psychosocial and physical health measures among Filipino and Filipino American adults (Flores *et al.*, 2018). Trained healthcare personnel administered interviewer-assisted questionnaires in local barangays, community centers and clinics. For the present analysis, only respondents who were at least 18 years old and residing in the Philippines at the time of data collection were included. Of the original 1,290 individuals surveyed, 1,196 met these criteria. Because respondents were recruited through convenience sampling, the sample may overrepresent certain demographic groups, especially older adults and residents of rural communities, a limitation noted in the study.

Participants were asked to identify their sources of stress in response to the question, "What causes your stress?" and to describe their stress management practices by answering, "Are there any other things that you do regularly to maintain your health and well-being?" Stressor options included work or school, family or childcare responsibilities, financial concerns and an open-ended "other" category. To support consistent interpretation across respondents, trained interviewers provided brief examples of each stressor category (e.g., work-related deadlines, childcare responsibilities and household financial obligations) to clarify participants' questions during survey administration. The original investigative team created the stressor and coping items used in the I-HELP-FILIPINOS survey to identify common sources of stress and everyday coping strategies within Filipino communities. They were informed by prior community health surveys conducted in the region. While not drawn from a single standardized instrument, the items reflect culturally salient experiences and were field-tested in similar low-income Filipino populations before full deployment. Because the items relied on self-report, some variation in how participants interpreted these categories may remain and is acknowledged as a study limitation.

Coping strategies encompassed eating, exercising, receiving massages, praying or participating in religious activities, staying at home, socializing and using acupuncture. Only strategies endorsed by at least 5% of the sample – such as eating, exercise, prayer, staying at home and massage – were included in the latent class analysis (LCA), because very low-frequency items, such as acupuncture, resulted in unstable model estimates. Psychological distress was measured with the validated Distress Thermometer (Bulli *et al.*, 2009), where higher scores indicate greater emotional distress. Sociodemographic variables, including age, sex at birth, marital status, education, income,

employment and rural *versus* urban residence, were used to explore group differences across emerging stressor–coping profiles.

Following data preparation, minimal missing responses (<1%) for key stressors or coping variables were noted, and listwise deletion was employed to satisfy the requirements of the poLCA mixture models package. The final analytic sample consisted of 1,193 participants. An analysis comparing individuals with and without missing data revealed no significant differences in age, gender or psychological distress, thereby indicating minimal bias resulting from item non-response.

### Ethical considerations

Data were de-identified before analysis. The study obtained ethics approval from the Institutional Review Board at the University of California, Irvine, and from the Research Ethics Board at the University of the Philippines Manila. Since this secondary analysis used anonymized data and involved minimal risk to participants, it did not require additional review. All procedures were conducted in accordance with ethical standards for research involving human subjects.

### Statistical analysis

After data processing, summary statistics of the demographic variables were calculated for the entire data set. Stressors and coping mechanism profiles were identified using latent class analysis (LCA). Latent class analysis employs Gaussian mixture models to identify hidden distinct classes within the target population, based on various outcome variables. An underlying assumption of the LCA is that membership in the identified classes can be explained by patterns in how participants responded to specific items (Weller et al., 2020). The Bayesian Information Criterion (BIC) and relative entropy were used to determine the appropriate number of classes for the LCA. The LCA was implemented using the poLCA (version 1.6.0.1) and poLCAExtra (version 0.1) packages in R (version 4.5.0) statistical software (Linzer and Lewis, 2011; R Core Team, 2021). A copy of the source code and output is available upon request. The solution with the lowest BIC, existing maximum likelihood estimates and estimated class membership share of >5% were considered. Interpretability of solutions and the relative entropy were also considered in the final class number selection.

To identify stressor and stress-coping profiles, the outcome variables included as predictors for class membership in the LCA were reported stressors (work, family care and financial concerns) and most commonly reported coping mechanisms (eating, exercising, staying at home, praying and receiving a massage). The samples were then classified into the identified classes by modal assignment using the class-conditional probabilities resulting from the LCA. Listwise deletion was implemented to account for missing responses. To ensure convergence of the LCA solution, 10 distinct initial points were used for maximum likelihood estimation with a maximum of 2,000 iterations.

After classification into the different profiles, associations between the profiles based on reported demographic variables and levels of psychological distress were examined post hoc. Associations between age and class membership were tested using the Kruskal–Wallis test. Associations between categorical demographic variables such as rurality of residence, income, employment and sex at birth were tested using the Mantel–Haenszel chi-squared tests. The strength of association between categorical variables was measured using Cramer's V ($\varphi_c$) adjusted for bias correction, while the rank eta-squared ($\eta_r^2$) was used to measure the strength of association involving age and psychological distress. For both measures, two-sided 90% confidence intervals are reported. A significance level of 0.05 was used for hypothesis testing.

## Results

### Demographics

Table 1 presents the demographics of the sample used in this study (N = 1,196). The sample was predominantly female (64.6%) with an average age of 49.6 years. The majority of participants were married (58.7%), earned less than PhP 50,000 annually (43.0%), had finished at least a high school education (77.4%) and were self-employed (21.2%) or homemakers (21.4%). Only 8.0% of the population resided in urban cities. These percentages suggest that the sample comprised rural residents who are educated and with diverse employment backgrounds and household compositions.

### Reported stressors

Figure 1a shows the causes of stress reported by the participants. 59.8% (N = 719) reported money as a stressor, followed by work or school (37.9%, N = 456) and family matters (34.5%, N = 415); 18.0% (N = 215) of the sample reported not applicable to the listed types of stressors. In the LCA, reports of unspecified types of stressors were excluded. This suggests that financial stress was reported as a dominant burden for the Filipino participants, regardless of employment and income level.

### Stress coping mechanisms

Figure 1b shows the reporting percentages of different stress management strategies in the sample. Eating (74.3%, N = 883), exercise (64.3%, N = 764) and prayer (57.5%, N = 684) were the top reported stress-relievers; 30.6% (N = 364) preferred to stay at home, and 15.0% (N = 179) reported getting massages to alleviate stress. There was a low percentage of the participants who reported using acupuncture to relieve stress (1.26%, N = 15), while 2.02% (N = 24) reported other unlisted methods. Acupuncture and other stress management strategies were excluded from the latent class analysis due to their low representation. These trends suggest that physical and spiritual nourishment were found to be preferred as stress relievers in this sample.

### LCA results: Profiles of stressors and coping mechanisms

There were three participants with missing responses, which were deleted during the implementation of the LCA, resulting in a final sample size of 1,193. The four-class solution yielded the lowest BIC (BIC = 11,630) and a relative entropy of 0.62, corresponding to an entropy $R^2$ value of 0.58 (Supplementary Table S1). Upon examining different solutions, the four-class solution configuration captured the most distinct and meaningful behavioral patterns across stressor types. While the corrected entropy value is lower than the typical threshold (0.8), other solutions with more classes had higher BIC values and non-convergence issues. Furthermore, expected class membership shares for all classes were all above 5% of the sample, which improves the interpretability of the solution. The Lo–Mendell–Rubin likelihood ratio test shows improvement from the three-class solution ($\Delta L = 44.5$, $\chi^2 = 84.9$, $p < 0.001$).

**Table 1.** Summary statistics of demographic variables for the overall data set (*N* = 1,196)

| Demographic | *N* (%) | Mean ± SD |
|---|---|---|
| **Age** | | 49.6 ± 17.7 |
| Missing | 1 | |
| **Assigned sex at birth** | | |
| Male | 424 (35.4%) | |
| Female | 772 (64.6%) | |
| **Location** | | |
| Urban | 93 (7.8%) | |
| Rural | 1,103 (92.2%) | |
| **Marital status** | | |
| Single | 270 (22.6%) | |
| Married | 700 (58.5%) | |
| Co-habitated | 2 (0.2%) | |
| Divorced/separated | 29 (2.4%) | |
| Widowed | 193 (16.1%) | |
| Missing | 2 (0.2%) | |
| **Education level** | | |
| No formal education | 2 (0.17%) | |
| Lower than elementary school | 46 (3.9%) | |
| Elementary school graduate | 150(12.5%) | |
| Lower than high school | 74 (6.2%) | |
| High school graduate | 323 (27.0%) | |
| Some college | 207 (17.3%) | |
| College graduate | 316 (26.4%) | |
| Postgraduate | 78 (6.5%) | |
| **Income** | | |
| < PHP 25,000 | 357 (29.8%) | |
| PHP 25 k-PHP 50 k | 323 (29.0%) | |
| PHP 50 k-PHP 75 k | 121 (10.1%) | |
| >PHP 75 k | 393 (32.9%) | |
| Missing | 2 (0.2%) | |
| **Employment** | | |
| Unemployed | 126 (10.5%) | |
| Government employee | 184 (15.4%) | |
| Non-government employee | 145 (12.1%) | |
| Self-employed | 252 (21.1%) | |
| Unpaid employment | 9 (0.8%) | |
| Student | 76 (6.4%) | |
| Homemaker | 254 (21.2%) | |
| Retired | 148 (12.4%) | |
| Missing | 2 (0.16%) | |

Figure 2a,b illustrates the class-conditional probability of class membership for the four profiles, categorized by types of stressors and coping mechanisms.

Class 1 (N = 238) describes individuals who are less likely to be affected by the stressors listed in the survey. Less than half of the participants are expected to attribute their stress to financial issues and are unlikely to cite work or family as a source of stress. Class 1 will be referred to as the "Mild Stress" (MS) class. Further investigation revealed that members of this class reported the highest percentage of other stressors (54.2%) compared to the different classes. Most participants in this class turn to exercise and eating habits to alleviate stress. Class 2 (N = 541) describes those who primarily attributed their stress to their place of work or school. These participants also chose a wide variety of stress-relieving practices, including massages and prayer. Class 2 will be referred to as the "Stressed Workers/Students" (SW) class. Class 3 (N = 209) describes those who were primarily stressed by family and financial matters and chose to relieve their stress through eating, exercising and praying. Class 3 will be referred to as the "Familial and Financial Stress" (FS) class. Unlike SW, FS did not report workplace or academic stress. Lastly, Class 4 (N = 205) describes those who were mostly stressed by financial matters and prefer to stay at home and eat to alleviate their stress. Class 4 was the only class that did not report exercise as a stress management practice. Class 4 will be referred to as the "Inactive Homebodies with Financial Stress" (IH) class.

### *Demographic disparities*

The breakdown of the demographic variables for each class is presented in Table 2. The Kruskal–Wallis test yielded strong evidence of association between class and age distribution ($\chi^2_3 = 217$, *p < 0.001*) with $\eta^2_r = 0.18$ (90% CI: 0.14–0.22). The Mantel–Haenszel chi-square test yielded strong evidence of association between class and sex at birth ($\varphi_c = 0.22$, 90% CI: 0.17–0.27, $\chi^2_3 = 61.0$, *p < 0.001*), income ($\varphi_c = 0.13$, 90% CI: 0.07–0.14, $\chi^2_{12} = 64.5$, *p < 0.001*), education ($\varphi_c = 0.29$, 90% CI: 0.25–0.31, $\chi^2_{21} = 328.4$, *p < 0.001*) and employment ($\varphi_c = 0.30$, 90% CI: 0.25–0.30, $\chi^2_{24} = 316.7$, *p < 0.001*). Younger participants were more likely to be classified into the SW profile, whereas older female participants were more likely to be classified into the high financial and familial stress profiles. MS had the lowest female representation (45.8%) among all classes and the highest percentage of retired participants (24.4%) within the group. SW recorded the lowest median age (39), which implies that SW comprised younger participants compared to the other classes. SW also had a higher percentage of singles (34.9%), college graduates (41.0%), employed participants (66.0%) and high-income earners (42.5% earned above PHP 75,000 annually). FS (41.2%) reported the highest percentage of high school graduates among all the classes, closely followed by MS (40.8%). FS also reported the highest percentage of homemakers (35.4%) and self-employed participants (26.8%). Meanwhile, IH reported the highest percentage of participants who had finished elementary/primary school (25.4%) and recorded the second-highest percentage of homemakers (34.2%).

### *Rurality of residence*

There was strong evidence of an association between rurality of residence and class ($\varphi_c = 0.16$, 90% CI: 0.11–0.21, $\chi^2_3 = 34.6$, *p < 0.001*). Participants from SW reported the highest percentage of urban residents (12.7%) out of all the behavioral profiles (Table 2). Additionally, 74.2% of the participants who reported living in an urban area were classified into Class 2. These differences suggest that individuals living in urban areas are more likely to worry about work-related matters. These findings support the hypothesis that

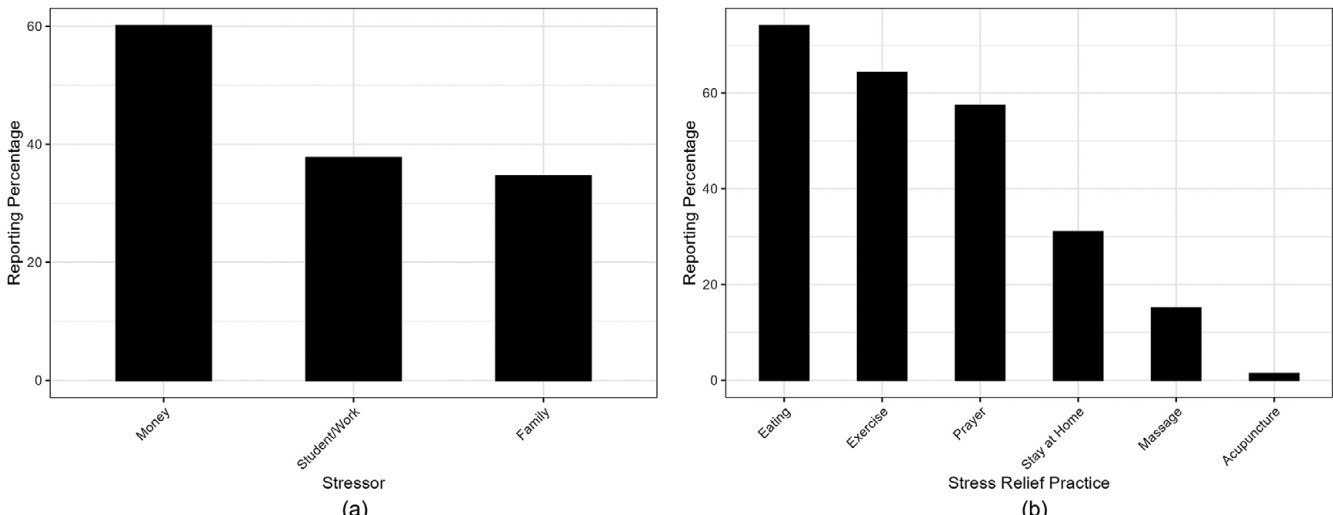

**Figure 1.** (a) The reporting percentages for each stressor type (*N* = 1,196). (b) The reporting percentages for each stress management strategy (*N* = 1,196).

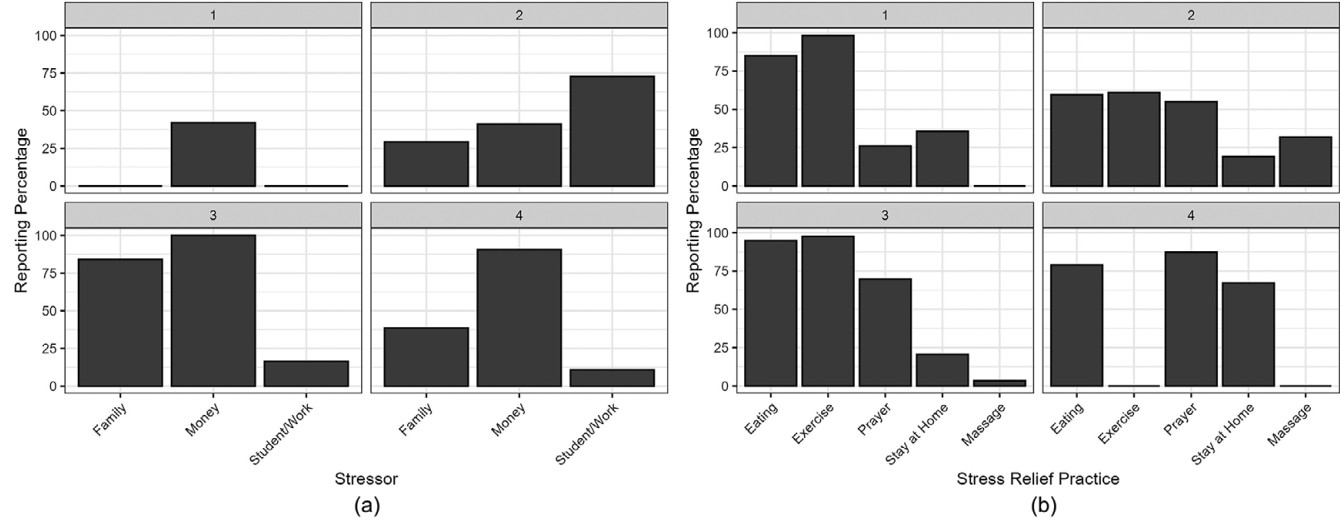

**Figure 2.** (a) The stressor profiles based on the four-class solution of the LCA (N = 1,196). (b) The profile of stress management strategies based on the four-class solution of the LCA (N = 1,196). Class 1: mild stress (MS), Class 2: stressed students/workers (SW), Class 3: familial and financial stress (FS), Class 4: inactive homebodies with financial stress (IH).

this behavioral profile is commonly experienced by young, career-oriented participants living in urban areas.

### Psychological distress

The Kruskal–Wallis test yielded strong evidence of an association between class and emotional distress ($\chi^2_3 = 217, p < 0.001$) with $\eta^2_r = 0.02$ (90% CI: 0.01–0.03). The overall median emotional distress score was 5 (IQR = 1) out of a maximum score of 10. FS (79.4%) and IH (70.2%) reported the highest percentage of participants scoring above the overall median emotional distress scores. Furthermore, FS recorded the highest median score with 5 (IQR = 0) while the other classes recorded a median score of 4, with the MS, SW and IH classes recording respective IQR values of 1, 2 and 1.

In general, MS comprised predominantly male retirees. SW is represented by younger, highly educated and employed urban individuals with high income. FS and IH consisted of older, predominantly female homemakers, with FS reporting higher levels of education,

income and psychological distress. These results highlight the difference in stress experiences and management methods across social demographic groups and rurality.

### Discussion

This study identified four profiles of stress causes and stress management strategies, based on reports from a sample of Filipinos living in the Philippines, categorizing participants according to their response patterns regarding stressors and stress management practices. The SW profile predominantly reported workplace stress, while the MS profile reported moderate financial stress. FS and IH reported high familial and financial stress, despite FS reporting income levels higher than MS. IH preferred to do at-home activities to cope, while participants with the FS profile preferred more social and nourishing activities. Sociodemographic disparities were observed between the classes, indicating the influence of

**Table 2.** Demographic Information for each class/profile ($N$ = 1,193)

| Demographic variable | Class 1 (MS) ($N$ = 238) | Class 2 (SW) ($N$ = 541) | Class 3 (FS) ($N$ = 209) | Class 4 (IH) ($N$ = 205) | Overall |
|---|---|---|---|---|---|
| **Age** | 57.1 ± 14.7 | 41.3 ± 16.6 | 54.4 ± 14.6 | 57.7 ± 16.9 | 49.6 ± 17.7 |
| **Assigned sex at birth** | | | | | |
| Male | 129 (54.2%) | 196 (36.2%) | 47 (22.5%) | 52 (25.4%) | 424 (35.4%) |
| Female | 109 (45.8%) | 345 (63.8%) | 162 (77.5%) | 153 (74.6%) | 772 (64.6%) |
| **Location** | | | | | |
| Urban | 6 (2.5%) | 69 (12.8%) | 10 (4.8%) | 8 (3.9%) | 93 (7.8%) |
| Rural | 232 (97.5%) | 472 (87.2%) | 199 (95.2%) | 197 (96.1%) | 1,103 (92.2%) |
| **Marital status** | | | | | |
| Single | 33 (13.9%) | 189 (34.9%) | 18 (8.6%) | 29 (14.2%) | 270 (22.6%) |
| Married | 153 (64.3%) | 298 (55.1%) | 141 (67.5%) | 106 (51.7%) | 700 (58.5%) |
| Co-habitated | 1 (0.4%) | 1 (0.2%) | 0 (0%) | 0 (0%) | 2 (0.2%) |
| Divorced/separated | 5 (2.1%) | 8 (1.5%) | 7 (3.4%) | 9 (4.4%) | 29 (2.4%) |
| Widowed | 46 (19.3%) | 13 (8.0%) | 43 (20.6%) | 61 (29.8%) | 193 (16.1%) |
| Missing | 0 (0%) | 2 (0.4%) | 0 (0%) | 0 (0%) | 2 (0.2%) |
| **Education level** | | | | | |
| No formal education | 1 (0.4%) | 0 (0%) | 0 (0%) | 1 (0.5%) | 2 (0.17%) |
| Lower than elementary school | 20 (8.4%) | 6 (1.1%) | 7 (3.4%) | 13 (6.3%) | 46 (3.9%) |
| Elementary school graduate | 41 (17.2%) | 19 (3.5%) | 38 (18.2%) | 52 (25.4%) | 150(12.5%) |
| Lower than high school | 22 (9.2%) | 17 (3.1%) | 14 (6.7%) | 21 (10.2%) | 74 (6.2%) |
| High school graduate | 97 (40.8%) | 84 (15.5%) | 86 (41.2%) | 55 (26.8%) | 323 (27.0%) |
| Some college | 23 (9.7%) | 128 (23.7%) | 27 (12.9%) | 29 (14.2%) | 207 (17.3%) |
| College graduate | 30 (12.6%) | 222 (41.0%) | 32 (15.3%) | 30 (14.6%) | 316 (26.4%) |
| Postgraduate | 4 (1.7%) | 65 (12.0%) | 5 (2.4%) | 4 (2.0%) | 78 (6.5%) |
| **Income** | | | | | |
| < PHP 25,000 | 97 (40.8%) | 130 (24.0%) | 63 (30.1%) | 66 (32.2%) | 357 (29.8%) |
| PHP 25 k–PHP 50 k | 62 (26.1%) | 141 (26.1%) | 53 (25.4%) | 65 (31.7%) | 323 (29.0%) |
| PHP 50 k–PHP 75 k | 25 (10.5%) | 39 (7.2%) | 32 (15.3%) | 25 (12.2%) | 121 (10.1%) |
| >PHP 75 k | 54 (22.7%) | 230 (42.5%) | 61 (29.2%) | 48 (23.4%) | 393 (32.9%) |
| Missing | 0 (0%) | 1 (0.2%) | 0 (0%) | 1 (0.5%) | 2 (0.2%) |
| **Employment** | | | | | |
| Unemployed | 36 (15.1%) | 40 (7.4%) | 23 (11.0%) | 27 (13.2%) | 126 (10.5%) |
| Government employee | 22 (9.2%) | 135 (25.0%) | 15 (7.2%) | 11 (5.4%) | 184 (15.4%) |
| Non-government employee | 15 (6.3%) | 107 (19.8%) | 11 (5.3%) | 11 (5.4%) | 145 (12.1%) |
| Self-employed | 40 (16.8%) | 115 (21.3%) | 56 (26.8%) | 41 (20.0%) | 252 (21.1%) |
| Unpaid employment | 2 (0.8%) | 4 (0.8%) | 0 (0%) | 3 (1.5%) | 9 (0.8%) |
| Student | 2 (0.8%) | 65 (12.0%) | 3 (1.4%) | 6 (2.9%) | 76 (6.4%) |
| Homemaker | 63 (26.5%) | 47 (8.7%) | 74 (35.4%) | 70 (34.2%) | 254 (21.2%) |
| Retired | 58 (24.4%) | 26 (4.8%) | 27 (12.9%) | 36 (17.6%) | 148 (12.4%) |
| Missing | 0 (0%) | 2 (0.4%) | 0 (0%) | 0 (0%) | 2 (0.16%) |

*Note:* Three participants were dropped from the LCA due to incomplete responses. FS, familial and financial stress; IH, inactive homebodies with financial stress; MS, monetary stress; SW, stressed workers/students.

social determinants of health on the causes of stress and coping mechanisms in the Filipino population. Differences in psychological distress were also observed between classes, indicating possible variations in stress responses across the patterns observed. The median distress score was the highest for FS, implying the need for targeted interventions involving stress management.

## Financial distress

Financial matters were the highest reported causes of stress in the MS, FS and IH classes. Filipino males were found to experience a lower occurrence of general stress and moderate financial stress (Sison *et al.*, 2014), as observed in the MS. MS was also found to have a high percentage of retirees, leading to a low likelihood of reporting employment-related stressors. FS cited family care issues, along with financial matters, as their top stressors. The high representation of self-employed participants and homemakers implies that the members of this class must maintain order in both work and home, which could indicate higher stress and psychological distress levels. Moreover, the relatively high income level for this profile could have been driven by the high percentage of self-employed participants. The results reflect that FS chose low-cost activities outside the home (eating, exercising, prayer), but shy away from the luxurious activities like massages to avoid incurring high costs. With the rising inflation rate in 2017 (de Vera, 2018), it might have been harder for FS to find stress relievers at the time of survey. This may have reflected in the elevated distress levels reported by the FS profile.

For IH, the high likelihood of reporting financial stress could reflect their choice of stress management. These participants were more likely to choose to stay at home, pray and eat to relieve their stress, all of which are associated with lower financial costs compared to outdoor activities such as massages and eating out at restaurants. The disinclination to exercise suggests that the participants may not have access to exercise equipment close to home, such as a home gym or at-home gym equipment, as this could also contribute to their current cost of living. This is in line with the transactional theory of stress and coping, as the secondary appraisal examines the degree of control or coping one has over the stressor (Lazarus and Folkman, 1987). Based on these findings, encouraging physical activity may not be as effective for members of the IH class in reducing stress levels.

## Interplay of stress and gender norms

Both FS and IH had a high proportion of females, which also suggests that gender norms in household and workplace settings may have contributed to the stress experienced by the participants in these classes. This expectation was found to be common among urban women in Southeast Asia (Nair, 2011). This was also reflected by the higher level of psychological distress in these classes compared to MS. The Filipino males' reported moderate financial stress appears to be in line with traditional gender role expectations, as Filipino men are more likely to have increased income-earning activities (Eder, 2006). This may create stressors related to bringing enough financial resources to provide for the household. Filipino women are more likely to perform domestic tasks that manage the household, including chores, allocation of financial resources and childcare. Household-related stressors may be partly dependent on the resources spouses bring home because of their role to ensure the home is appropriately managed (Akter et al., 2017; Torell et al., 2021). Unlike FS, IH did not report family as a leading cause of stress, despite reporting similar levels of psychological distress. The low tension between family members could have helped build an environment conducive to relieving stress at home for IH.

## Work-related stress

SW reporting higher probabilities of work-related stressors was expected due to the high employment rate in the group. It was also not surprising that the members of this class recorded the lowest median age across the classes, as Class 2 had low retiree representation (4.81%).

The attitudes of the worker and student population toward stress have been the most studied in prior literature. The stress responses were found to be more affective and behavioral in Filipino students (Dy et al., 2015b). Students are likely to encounter stressors related to their academic performance, such as excessive course load, career indecision and expectations from family and professors (Flores-Buils and Mateu-Pérez, 2025). Previous research has suggested that Filipino university students utilize coping skills that effectively distract them from their stressors and promote positive affect, which may contribute to the range of coping skills observed in the current study (Bate et al., 2019). The moderate level of familial stress for the SW class contrasts with existing literature that suggested that academic pressure could carry over to familial stress among Filipino students due to culturally specific familial dynamics (Dizon et al., 2025).

For workers, previous research has suggested that there are different degrees of stress among professionals at various career stages. Those in their late career stages tend to report less work-related stress, suggesting that stress management strategies should be tailored to each career stage (Yasmeen and Supriya, 2010; Dorociak et al., 2017).

Previous research also identified differences in stress appraisal between age cohorts. In particular, older individuals were found to be less reactive to potential stressors (Hechanova et al., 2022). It may be that late career individuals are more likely to appraise stressors more accurately by differentiating what is an actual threat from a challenge or benefit (Lazarus and Folkman, 1987; Hechanova et al., 2022). As SW had the lowest median age, the variety of stress management strategies in this class may reflect the nature, demands, roles and responsibilities of one's occupation at the early-to-mid-career stage.

The SW class might find activities involving physical and spiritual nourishment as appealing stress-management practices. The low preference for staying at home reflects how the SW class prefer the use of third spaces to relieve their stress. Unlike the FS class, the SW class might consider luxurious activities like massages from time to time for stress relief.

## Prayer as stress management

Turning to religion and food are historically common methods of relieving stress. Both religion and food are culturally embedded forms of coping with stress among Filipinos, owing to the country's rich religious and culinary heritage. Depending on the region, Roman Catholic and Muslim faiths may be inextricably embedded in the Filipino culture (Alviar and del Prado, 2022). Religiosity, which includes prayer and access to community, was identified as an important domain of coping for Filipinos (Rilveria, 2018). Positive religious coping was found to be an essential buffer against the negative impact of daily stressors in Filipinos (Bulisig and Aruta, 2023). In addition, Filipinos were found more likely to turn to religious clergy prior to seeking formal help when distressed (Martinez et al., 2020). Spirituality was one of the top stress-coping strategies used by Filipino college students in rural areas, which was consistent with the cross-conditional probabilities for SW (Nuas et al., 2015; Austria-Cruz, 2019). Utilizing religious resources may be a particularly important coping strategy, as an updated transactional theory of stress

and coping also suggest that individuals facing uncontrollable or overwhelming stressors may adaptively cope by turning to their beliefs and values to find meaning in their distress (Biggs et al., 2017). Dispositional factors can impact appraisals, and spirituality and religious involvement may be important factors in enhancing dispositional resilience (Gusilatar et al., 2025).

On the other hand, MS had the lowest prayer utilization among the classes, which contradicted prior research that reported religion as one of the top mechanisms for male-majority agricultural cohorts (Ambong and Gonzales, 2019). The findings imply that stress management practices rooted in prayer may not be as appealing to older male Filipinos who belong to the MS class. Furthermore, the IH class might find reflective spiritual activities like meditation appealing as it could be done within the confines of one's home.

### Food in stress management

All classes reported eating as a stress management strategy. Stress can alter one's food selection by causing individuals to choose higher-volume, higher-calorie foods that are more palatable, with the intent of reducing their stress response (Ulrich-Lai et al., 2015). Despite how stress responses may include decreased or unchanged food intake, turning to food consumption as a form of coping is still notable because of how eating may alter or ameliorate one's negative affect as a result of stress. Combined with the collectivist nature of Filipinos (King et al., 2014), these results imply that social support through food, such as communal dining, could help Filipinos cope with stressful times. The preference on the setting of communal dining may be reflected by the preferences of the other stress-relieving practices. For example, the IH class might prefer to host dinners at home compared to FS who might prefer to eat at third spaces with friends and community members.

### Strengths

The data set used in this study has a large sample size from Filipinos of diverse backgrounds, providing insight on the stress-related behavior of a more general Filipino population. Using LCA to identify stressors and stress management patterns underscores the study's novelty, as it provides a granular, person-centered analysis of stress that is largely underutilized in the Filipino mental health literature.

### Limitations

The study design employed was cross-sectional; therefore, causal conclusions cannot be drawn from this study. Although the sample size is more than adequate for latent class analysis, all recruited participants were from Northern Luzon. Thus, the results might not be generalizable to all Filipinos in the Philippines as convenience sampling was used to recruit participants. The choice selection for the stressors and stress management practice was not exhaustive and might not have accounted for all relevant options for the survey participants. Lastly, the information received from the participants was self-reported and stressor categories could be interpreted differently by participants, which could lead to potentially biased measurements.

### Conclusions

Profiles of stress causes and management strategies were found to be associated with sociocultural factors in the Philippines. The existence of the different profiles suggests that a one-size-fits-all intervention to promote better stress management would not be effective for certain individuals, especially in marginalized populations defined by rurality, age and income. Profiles associated with higher psychological distress were also identified. This study addresses the gap in understanding the causes of stress and stress management among rural and elderly Filipinos. The identified profiles provide a foundation for the development of evidence-based interventions that are culturally tailored to Filipinos, especially rural and older adults.

The implementation of a nationwide post-pandemic survey, as well as a qualitative research on stress management exploring the profiles discovered in this study, is recommended for future studies, as more stressors may have arisen due to the COVID-19 pandemic. Extending the study to Filipino immigrants could also provide further insight into how Filipinos worldwide experience and cope with stress in diverse sociocultural environments. Temporal transitions between discovered profiles could be explored by extending the LCA to longitudinal cohort studies. Lastly, identifying mediators and moderators between class membership and other health outcomes through a longitudinal study is recommended for future studies.

**Open peer review.** To view the open peer review materials for this article, please visit http://doi.org/10.1017/gmh.2026.10192.

**Supplementary material.** The supplementary material for this article can be found at http://doi.org/10.1017/gmh.2026.10192.

**Data availability statement.** The data that support the findings of this study are available on request from the senior author, LSE. The data are not publicly available due to containing information that could compromise the privacy of research participants.

**Author contribution.** Conceptualization: M.A.F., P.P.P., M.V.; Data Curation: M.A.F., L.S.E.; Formal Analysis: M.A.F.; Investigation: M.A.F., P.P.P., M.V.; Methodology: M.A.F., L.S.E.; Project Administration: L.S.E.; Resources: L.S.E.; Software: M.A.F.; Supervision: M.A.F., L.S.E.; Visualization: M.A.F.; Writing – Original Draft: M.A.F., P.P.P., M.V., L.S.E.; Writing – Review and Editing: M.A.F., P.P.P., M.V., L.S.E.

**Financial support.** This research received no specific grant from any funding agency, commercial or not-for-profit sectors.

**Competing interests.** All authors have agreed to the submission, and there are no conflicts of interest to disclose.

**AI use.** AI tools were used in the writing process for this manuscript. Google Gemini 2.5 Flash was used for proofreading purposes only.

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
