## [Reviewer Report]

This study informs us how Filipinos experience and manage stress. By analyzing responses from over 1,196 people, the researchers identified four distinct stress profiles. Additionally, the latent class analysis was great as this examined how stress and coping strategies vary depending on people’s backgrounds, showing that culture and social factors shape how stress is felt and managed.

Although, here are some suggestions for improvement:

1. In the discussion, consider breaking it with shorter paragraphs and use of subheadings. This would make it easier to follow.

2. Simplifying complex sentences in the discussion will improve readability.

3. The discussion about gender is insightful however, the authors may provide more focused explanation of how these “norms” directly impact their stress and coping.

4. A brief discussion on connecting these coping strategies to intervention design would be good. A discussion on the insights on these coping strategies per classes can inform culturally sensitive and appropriate mental health programs.

Overall, this study gives us insightful findings that may be used for designing mental health programs that fit the needs of the Filipino people.

---

## [Reviewer Report]

This is an interesting and timely study that applies latent class analysis to identify behavioral patterns related to stressors and coping strategies among a Filipino sample. The topic is highly relevant, particularly given the limited research on stress and coping among populations in Southeast Asia. The analytical approach is appropriate for uncovering underlying heterogeneity in coping responses, and the manuscript makes a potentially valuable contribution to understanding culturally specific stress responses in the Philippines.

However, a few points require clarification and refinement to strengthen the paper’s methodological rigor and interpretive clarity. Below are my specific comments and recommendations.

Major Comments:

Relevance of Data to Current Context

The data were collected in 2017, and much has changed since then in terms of socio-economic, environmental, and public health stressors in the Philippines (e.g., COVID-19, political shifts, climate events). The authors should comment on the relevance of their findings in light of these recent developments, and whether the identified latent classes might still be applicable today.

Clarification of Sample Composition

Although the paper states that all respondents resided in the Philippines at the time of the survey, it would be helpful to clarify the proportion of participants who were Filipino versus Filipino American. This information is important for contextualizing cultural influences on stress and coping patterns.

Handling of Missing Data

The authors note that listwise deletion was used, but there is no summary of the extent of missingness or its potential impact on the analytic sample. Please include a brief description in the Methods section (e.g., proportion of cases excluded and any differences between included/excluded participants, if examined).

Interpretation and Labeling of Latent Classes

Rather than referring to classes generically as “Class 1,” “Class 2,” etc., the authors should consider assigning descriptive labels that capture the defining features of each class. This will make the findings more interpretable and communicable to readers and practitioners.

Presentation of Sociodemographic Associations (Table 1)

There appear to be two tables labeled Table 1; the second should likely be Table 2. I recommend including row percentages (rather than or in addition to column percentages) in this table (Table 2) to highlight which latent classes most likely characterize each sociodemographic groups. This approach provides more actionable insight for intervention design, as it identifies which specific stress-coping profiles describe and mostly likely going to work with specific demographic groups. We generally know in advance the sociodemographic groups we design public health interventions for.

Quantifying Disparities Between Demographic Groups

The reliance on simple statistical tests (e.g., chi-square and p-values) limits the interpretation of effect sizes. Consider using logistic or log-binomial regression to quantify associations (magnitude of disparities) between sociodemographic factors and latent class membership. Reporting odds ratios or prevalence ratios with 95% confidence intervals will help communicate both the magnitude and precision of observed disparities. Some of the demographic groupings may need to be collapsed due to small cell sizes.

Inconsistency in Analytical Approach

It is unclear why the analysis of rural/urban residence was performed using regression, whereas other sociodemographic variables were tested with bivariate methods. Please clarify the rationale for this analytical decision and consider applying a consistent approach across variables.

Limitations Related to Item Selection

The authors should note that the study’s findings are constrained by the specific stressors and coping strategies included in and at the time of the survey. Other relevant items not captured could have led to different latent class solutions. This limitation should be explicitly acknowledged in the Discussion section.

Lastly, consider expanding the Discussion to reflect implications for intervention design, especially if certain coping profiles are more common in specific demographic groups.

---

## [Reviewer Report]

Abstract

1. You might consider emphasizing the analytical approach earlier to highlight the novelty (e.g., “Using latent class analysis, we identified…” before listing results).

2. The Study Design statement could be shortened for smoother flow.

Suggest adding one quantitative indicator (e.g., “Four stress-related profiles explained X% of variation”) to increase precision.

3. The conclusion could briefly hint at potential applications (e.g., “These findings can guide targeted mental health interventions in LMIC settings”).

Impact Statement

1. Nicely contextualized within Filipino culture and global mental health, but it may benefit from a stronger link to policy or practice implications.

2. Some sentences are quite long; shortening them would improve readability.

3. You could add one sentence explicitly stating how the findings extend or challenge existing literature.

4. Consider softening speculative claims (e.g., “provides evidence-based guidelines” → “could inform evidence-based guidelines”).

Introduction

1. Well-researched and comprehensive, but it could feel tighter. The long paragraph on cardiovascular disease (IHD, CHD, stroke) slightly pulls attention away from your main goal, stress profiling. Maybe condense those stats into one sentence to maintain focus on mental health and stress.

2. You’re citing foundational works (Lazarus 1984, McEwen 1993), which are great for theory, but add a few newer studies (post-2020) on stress frameworks or allostatic load to make the argument feel more current.

3. Explain why LCA fits the question. Readers unfamiliar with latent class analysis might wonder why it’s used instead of simple clustering or regression. Add one short line explaining that LCA identifies hidden subgroups and allows for probabilistic classification, making it ideal for examining stress–coping patterns.

4. Sharpen the knowledge gap. The “urban vs. rural” gap is a start, but be specific, say that prior Filipino stress studies mostly focused on students and health workers, not general community populations. Highlight that no study has yet mapped distinct stressor–coping typologies using a national sample.

5. Smoother flow between global and local context. Some paragraphs feel like abrupt jumps from international data to the Philippine situation. Use linking phrases (“Building on global findings,” “In contrast, among Filipinos…”) to make transitions more natural.

6. Align better with the journal’s focus. Since Cambridge Prisms: Global Mental Health values cross-cultural insights, end the intro by situating your study in the broader Southeast Asian or LMIC stress literature. One bridging line would make this stronger.

7. Break up long sentences and use transitions like “Moreover,” “Furthermore,” “To address this gap” to make it more readable.

Methods

1. Good clarity on dataset source, but the reference to the parent study (I-HELP-FILIPINOS) could be expanded with a sentence on its original objectives, data collection period, and representativeness. This helps readers unfamiliar with the parent dataset understand its scope and limitations.

2. Sampling approach needs more details. Since “convenience sampling” limits generalizability, it would be useful to acknowledge this upfront and discuss possible biases (e.g., overrepresentation of certain regions, education levels, or occupations). You can also note any strengths that mitigate this.

3. Clarify exclusion and inclusion of variables. Some stressors and coping mechanisms were excluded for low frequencies (like acupuncture). Briefly justify this decision. Mention thresholds (e.g., <5% prevalence) or analytical reasons (instability of estimates in small categories).

4. Statistical Analysis section is okay but can be more transparent. Explain why latent class analysis was preferred over other techniques, mention it allows for discovery of unobserved subgroups and models categorical variables without assuming linearity.

5. Model fit evaluation could be expanded. BIC and entropy are mentioned, but consider stating if alternative fit indices (AIC, likelihood ratio test) were compared or if interpretability guided the final class selection.

6. Add one or two lines on software reproducibility. You already mentioned using R and poLCA; including the version or package citation (e.g., “R version 4.3.2, poLCA 1.4.1”) supports transparency and replicability.

7. Clarify how covariates were treated. Indicate if demographic variables were used as predictors of class membership post hoc or as part of the model.

8. Missing data handling section deserves a more expanded version. “Listwise deletion” is mentioned—briefly acknowledge potential data loss and whether sensitivity analyses were conducted to ensure results remained robust.

9. The ethics approval details are clear but lengthy; move them under a new subheading (“Ethical Considerations”) to align with standard journal formatting. Summarize in one sentence (e.g., “The protocol was approved by UC Irvine IRB and UPMREB).

10. Consider mentioning data access and anonymity. Since this is secondary analysis, add one clarifying line that the dataset was de-identified before analysis, ensuring compliance with data privacy standards.

11. Overall, the Methods section reads well but could use short transition phrases between subsections (e.g., “Following data preparation,” “To identify stressor–coping profiles,” “Subsequently, associations were tested”).

Results.

1. Clear presentation overall, but the narrative feels too segmented. Try weaving short interpretive comments after each table or figure; for example, briefly summarizing what the pattern means rather than jumping straight to the next dataset.

2. Consider stronger storytelling. Right now, it reads as “Table–Figure–Result.” Add short bridging sentences like, “This suggests that financial stress remains a dominant burden even among middle-income participants.”

3. About LCA, you could add one line on interpretability; why the 4-class model made the most sense theoretically or contextually (e.g., “This configuration captured the most distinct and meaningful behavioral patterns across stressor types”).

4. Include additional fit statistics if available. Mentioning AIC, adjusted BIC, or likelihood ratio tests (if run) would further support the choice of the 4-class model.

5. Figure captions should be more self-contained. State the total sample (N=1,196) and clarify that values are percentages or proportions. Readers should understand the figure without reading the main text.

6. Add short comparative remarks. For example, after describing each class, summarize how they differ (“Compared to Class 2, Classes 3 and 4 showed higher distress levels and were more rural”). These mini-summaries help readers follow patterns quickly.

7. Demographic comparison table is detailed, but it can be followed by one integrating sentence. Something like: “Overall, Class 2 represented younger, urban, employed individuals, while Classes 3 and 4 were older and more rural, with homemaking or self-employment as main occupations.”

8. Psychological distress subsection is good, but include more statistics to give depth: report median or mean scores, interquartile ranges, or simple effect sizes instead of only chi-square results. It’ll make the findings feel more substantial.

9. Highlight key associations visually or narratively. You could add a sentence linking demographics to stressor types (e.g., “Older participants tended to cluster in financially stressed profiles, whereas younger ones fell under work-related classes”).

10. Add a short synthesis paragraph at the end. A brief “Summary of Findings” line helps transition into the Discussion; something like: “These results demonstrate that stress experiences among Filipinos cluster distinctly by socioeconomic role, gender, and rurality.”

11. You could also reduce redundancy. If any table values are repeated verbatim in the text, summarize trends instead of listing numbers again. This keeps the section concise and readable.

Discussion

1. This section can flow better. Start the section with one concise overview sentence summarizing the four identified classes and their defining stress; coping profiles before moving into interpretations. This orients readers right away.

2. Interpretation is rich, but some paragraphs read like restatements of the Results. Focus more on why these patterns exist and what they mean in relation to Filipino sociocultural contexts (e.g., collectivism, religiosity, gender norms).

3. Could you deepen the theoretical linkage? Explicitly tie findings back to the Lazarus stress appraisal model or other stress frameworks mentioned earlier. This closes the conceptual loop and strengthens theoretical grounding.

4. Highlight what’s new. You could stress that this is one of the few studies using latent class analysis to map Filipino stress profiles, making it a methodological and contextual contribution. Mention this novelty clearly.

5. Cultural interpretations are strong, especially about gender and prayer, but you could support them with one or two local citations (e.g., studies on Filipino religiosity, gendered care roles, or rural coping styles) to anchor them in Philippine literature.

6. Be mindful of paragraph length. Some sections (like gender norms and rural context) are long. Consider breaking them into smaller, focused paragraphs; each addressing one major insight (e.g., “financial stress and gender,” “religious coping,” “urban vs rural contrasts”).

7. Discuss practical implications earlier. Don’t wait until the end to connect findings to mental health policy or intervention design. Insert brief applied reflections throughout; e.g., “This finding suggests that interventions in rural settings may benefit from integrating community prayer or group-based coping strategies.”

8. Strengthen linkage between class patterns and distress levels. Highlight that Classes 3 and 4 not only reported more financial/familial stress but also showed higher distress scores, underscoring the need for targeted psychosocial support.

9. One paragraph can tie the Filipino stress experience to global South contexts, how sociocultural and economic structures shape stress differently from Western populations. This fits the journal’s “Global Mental Health” scope.

10. Could you limit overinterpretation? A few statements risk sounding causal (e.g., “financial stress could have been related to their choice of stress management”). Soften these with phrasing like “may reflect” or “could indicate.”

11. Add one or two quantitative anchors. When describing class differences, include approximate percentages or direction of association (e.g., “Class 2 participants were mostly younger and employed, aligning with the lower distress scores observed”).

12. Strengths and limitations discussion is brief. Strengthen it by noting that (1) the large community-based dataset gives population insight; (2) the cross-sectional nature and convenience sampling limit generalizability; and (3) self-report may introduce bias.

13. Future directions could be clearer. Suggest how LCA could be applied longitudinally or combined with qualitative inquiry to understand evolving stress responses, especially post-pandemic.

14. End with a forward-looking, integrative line, something like: “By identifying distinct stress profiles, this study provides an evidence base for designing culturally responsive stress management programs in the Philippines.”

---

## [Reviewer Report]

This is an interesting study. A few points that the authors can take into consideration -

1. Page 4 line 30-46. In the introduction, the authors mentioned cardiovascular disease and its association with stress, but the actual analysis mentioned nothing about cardiovascular disease in the dataset. It would be better to be more precise as to why this part was included in the introduction.

2. Page 4 line 18-31. Similar to point 1. Academic pressure seems not to be a variable in the data, or it is, but with a relatively small portion reporting having academic pressure. Please be more precise regarding the connection between the introduction and your data, why this factor is important, and how it relates to the data chosen for this study.

3. Page 5, line 48. The importance of this study is to “ enhance public health interventions in the country”, yet the data utilised were regional (i.e., Northern Luzon, see page 13 line 54). As this result may not be generalised to the national level, please make sure that the contribution of this study at either the introduction or limitations is consistent.

4. Whether the participants given the definitions or examples of the stressor? Could it be possible that they had different interpretations of what stress might look like?

5. Please provide more details regarding the data used in the study. What were the instruments, who and when was it established? What was the purpose of these instruments? Was it a subscale from a larger survey?

6. Does not necessarily need to carry out another analysis. But whether the authors controlled the mediators and/or moderators when performing statistical analysis? For instance, the age to stress levels, gender to the stressors and coping mechanisms. It would be great to include these in the methods for more accurate results on the stressors and coping strategies in the chosen data.

6.1-- Class 4, please consider that gender and education may be mediators of the stress and coping strategy. For example, females were the main population of this class. Would it be possible that women (lower educational attainment) were more likely to stay at home at the time, causing higher financial problems, and higher financial problems causing higher levels of emotional distress, higher emotional distress then cause them to stay at home and such.

7. If the data were collected from Northern Luzon, what was the average income and living cost at the time when the data were collected? Compared to the class reported having financial problems, whether their finance were significantly lower compared to the average?

---

## [Editor Report]

Thank you for submitting your manuscript, “Identifying Profiles of Stressors and Stress Management Strategies in Filipinos: A Secondary Analysis” to Cambridge Prisms: Global Mental Health.

All reviewers have recognized the importance of your study; however, they have also noted some inconsistencies and unclear sections especially in the methodology and discussion section. You are requested to revise your manuscript accordingly, addressing the reviewers’ comments, and resubmit it for further consideration.

---

## [Reviewer Report]

The authors have clearly taken the previous comments seriously, and the manuscript is much stronger as a result. The revisions improve both clarity and coherence across sections, especially in the abstract and introduction, which now better highlight the use and value of latent class analysis and situate the study within more current and relevant literature. The Methods section is more transparent and reader-friendly, with clearer explanations of the dataset, sampling approach, model selection, missing data, and reproducibility. The Results now read more like a story than a sequence of tables, with helpful transitions and added quantitative detail, while the Discussion opens more effectively, connects findings to theory and Filipino sociocultural contexts, and weaves in practical implications throughout. Overall, the paper is easier to follow, more rigorous, and more compelling, and the authors should be commended for thoughtfully engaging with the feedback. Congratulations!

---

## [Reviewer Report]

Thank you for the opportunity to review your revised manuscript. I commend you for the thoughtful and thorough way in which you have addressed the reviewer comments. The paper has clearly improved in terms of clarity, structure, and overall coherence, and I believe it is close to being suitable for publication.

My primary remaining concern relates to the interpretation and reporting of statistical results, specifically the continued reliance on p-values to support statements about the “strength” of associations. For example, on page 33, lines 311-314, the authors wrote, "The Mantel-Haenszel chi-square test yielded strong evidence of significant

association based on the Mantel-Haenszel chi-square test between class and class and age distribution (2=217, p<0.001), sex at birth (2= 61.0, p<0.001), income (2

=64.5, p<0.001), education (2=328.4, p<0.001), and employment (2=316.7, p<0.001)."

While statistical significance can be informative, it can also be misleading, particularly in large samples, where even trivial or non-meaningful effects may achieve statistical significance. As such, framing inferences about the strength of associations primarily on p-values risks overstating the substantive importance of the findings.

I encourage you to revise the Results section so that statements about the magnitude or strength of associations are grounded primarily in the estimated effect sizes (odds ratio or prevalence ratio) and their associated measures of precision (e.g., 95% confidence intervals), rather than statistical significance alone. Reporting and interpreting the size of effects alongside their uncertainty will improve the transparency and interpretability of the findings for readers.

For guidance on best practices in statistical reporting, you may find the American Statistical Association’s statement on p-values particularly helpful:

https://www.amstat.org/asa/files/pdfs/p-valuestatement.pdf

With this relatively minor revision, I believe the manuscript would be well-positioned for acceptance. I appreciate the care and effort you have invested in this work and look forward to seeing the revised version.

---

## [Editor Report]

Thank you for submitting your revised manuscript “Identifying Profiles of Stressors and Stress Management Strategies in Filipinos: A Secondary Analysis”. The reviewers have noted your revisions and found the paper significantly improved. Please address the minor revision outlined by the reviewer below regarding the reporting in the Results section.

---

## [Editor Report]

Thank you for submitting your revision of the manuscript titled: Identifying Profiles of Stressors and Stress Management Strategies in Filipinos: A Secondary Analysis. The revisions successfully address all points raised during the review process, and the manuscript is now suitable for publication in its current form